# OGGSPLAT: OPEN-VOCABULARY GAUSSIAN GROWING FOR EXPANDED FIELD-OF-VIEW

## ABSTRACT

Reconstructing open-vocabulary 3D scenes from sparse views is both challenging and crucial, driven by the demands of emerging applications such as virtual reality and embodied AI. However, existing generalizable open-vocabulary 3D Gaussian Splatting methods struggle to reconstruct semantically enriched regions outside the input view cone. To address this limitation, we introduce OGGSplat, an open-vocabulary Gaussian growing method that extends the field-of-view for generalizable, semantically-enriched 3D scene reconstruction. Our key insight is that the semantic attributes of open-vocabulary Gaussians serve as strong priors for image extrapolation, ensuring both semantic consistency and visual plausibility. Specifically, once Gaussians with semantic attributes are initialized from sparse views, we introduce an RGB-semantic consistent inpainting module to selected rendered views. This module enables bidirectional control between an image diffusion model and a semantic diffusion model. The inpainted regions integrated with semantics are then lifted back into 3D space for efficient, progressive optimization of Gaussian parameters. To evaluate our method, we propose the Open-Vocabulary Gaussian Outpainting (OVGO) benchmark, which measures both the semantic and generative quality of the reconstructed open-vocabulary scenes. OG-GSplat also demonstrates promising semantic-aware reconstruction capabilities when provided with two views captured directly from a smartphone camera.

## 1 INTRODUCTION

Building realistic and semantically meaningful 3D representations of the world has become a crucial goal in computer vision, driven by applications in robotics, virtual reality, and embodied AI. Beyond reconstructing vivid textures and accurate geometry, modern systems increasingly demand semantic awareness to support high-level understanding and interaction within 3D environments. This dual demand for geometric fidelity and semantic interpretability introduces new challenges for scene representation. Recent researches typically address this by combining open-vocabulary features with 3D reconstructive representations like 3D Gaussians (Kerbl et al., 2023). Approaches based on per-scene optimization (Qin et al., 2024; Shi et al., 2024; Qu et al., 2024; Qiu et al., 2024; Wu et al., 2024; Ye et al., 2024), which leverage dense multi-view inputs, achieve well-structured 3D geometry with fine-grained semantic alignment. In contrast, newly emerging feed-forward methods (Wang et al., 2024b; Hu et al., 2024) offer improved scalability and generalization across scenes by predicting semantic-aware 3D representations directly from sparse input views via a trained neural network.

Although recent feed-forward open-vocabulary 3D Gaussian reconstruction methods enable fast inference and efficiently handle sparse input views, their performance is often constrained by the limited scope of sparse inputs. For extremely sparse views, such as those with only two input perspectives, the semantic-aware 3D reconstructions can exhibit distorted geometry and semantically implausible content when extrapolated to novel viewpoints. For example, in Figure 1, the black regions in the novel view arise from the absence of Gaussians, resulting in incomplete reconstructions for both 3D scene and semantics. This highlights an urgent need for a generalizable open-vocabulary 3D reconstruction framework that can reliably expand the field-of-view while maintaining geometric coherence and semantic consistency. We argue that incorporating semantic cues from open-vocabulary features can provide valuable guidance in imagining plausible content for unseen regions, thus extending the application of generalizable reconstruction.

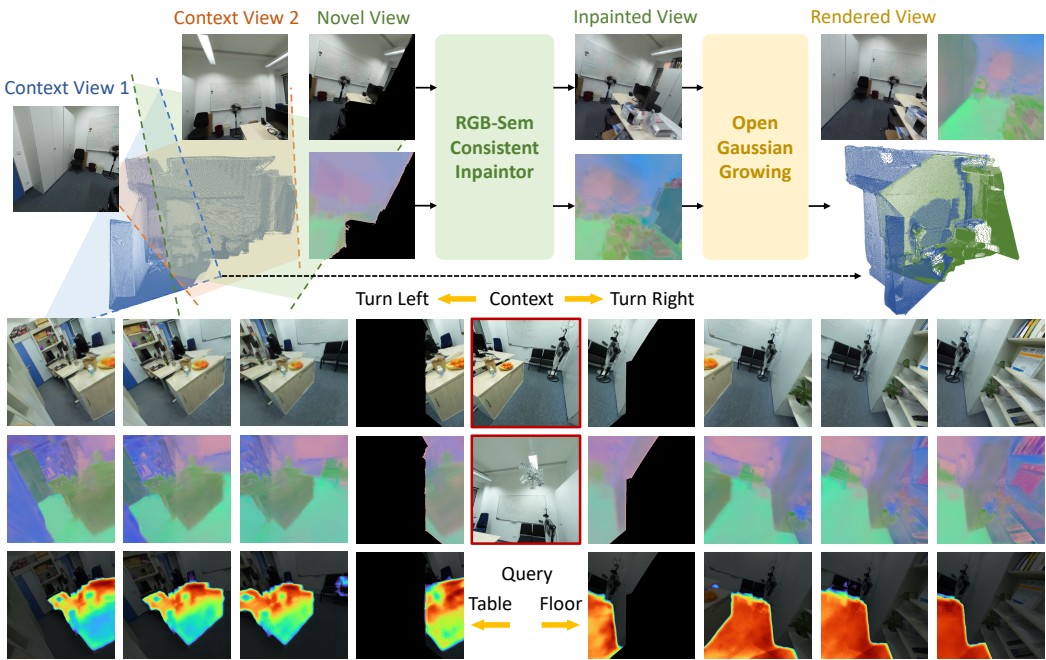

Figure 1: We propose **OGGSplat**, an open-vocabulary Gaussian growing method that expands the field-of-view of semantic-aware generalizable Gaussian reconstruction. The last three rows visualize the rendered images, semantic maps, and heatmaps obtained through open-vocabulary queries.

In this paper, we address the aforementioned challenge of generalizable open-vocabulary 3D reconstruction by introducing **OGGSplat**, an **O**pen-Vocabulary **G**aussian **G**rowing framework designed to extrapolate semantically meaningful 3D Gaussians beyond the input view coverage. Our goal is to enhance open-vocabulary Gaussian representations with the capacity to grow new, semantic-aware Gaussians, thereby recovering occluded parts or expanding the field-of-view in open-vocabulary scenes reconstructed from sparse inputs. A key insight of our approach is that the semantic attributes inherent in open-vocabulary Gaussians provide a strong prior for semantically plausible extrapolation. To exploit this, OGGSplat uses a progressive Gaussian growing strategy that builds on the initial reconstruction from sparse views. Central to this process is a novel RGB-semantic consistent inpainting module, which enables bidirectional interaction between image and semantic inpainting: semantic maps guide image completion, while inpainted images refine the semantic features in return, ensuring pixel-level alignment. The synthesized RGB images and semantic maps are then used to efficiently optimize the newly introduced Gaussians.

We conduct extensive experiments on ScanNet++ (Yeshwanth et al., 2023) and introduce a novel Open-Vocabulary Gaussian Outpainting (OVGO) benchmark. OVGO evaluates both visual fidelity and semantic plausibility in extrapolated regions, enabling quantitative assessment via segmentation mean Intersection-over-Union (mIoU) alongside generative metrics such as Fréchet Inception Distance (FID) (Heusel et al., 2017). We further demonstrate OGGSplat on context images captured with a smartphone, highlighting its potential for portable applications.

In conclusion, the contributions can be summarized as: (1) We propose OGGSplat, the first work to expand the field-of-view for generalizable open-vocabulary Gaussian reconstructions. (2) We design an RGB-semantic consistent inpainting module that enforces bidirectional interaction between image and semantic map inpainting, and introduce a progressive Gaussian growing strategy to optimize new Gaussians from the inpainted content. (3) We establish the Open-Vocabulary Gaussian Outpainting (OVGO) benchmark, enabling semantic-aware evaluation on expanded regions.

## 2 RELATED WORK

**3D Gaussian Splatting.** Existing 3D Gaussian Splatting (3DGS) approaches can be broadly categorized by their optimization strategy and the number of input views. Early methods (Yu et al.,

2024; Lu et al., 2024; Fan et al., 2024a; Fu et al., 2024) rely on per-scene optimization with hundreds of images, achieving high-fidelity reconstructions but suffering from high computational cost and limited scalability. Subsequent works (Xiong et al., 2023; Paliwal et al., 2024; Chung et al., 2024; Zhu et al., 2024) aim to reconstruct scenes from only a few input views. More recently, generalizable 3D reconstruction methods such as PixelSplat (Charatan et al., 2024) and MVSplat (Chen et al., 2024) have been proposed to avoid per-scene optimization by leveraging feed-forward neural networks trained on large-scale datasets. Splatt3R (Smart et al., 2024) directly infers point clouds and Gaussian parameters from unposed image pairs, eliminating the need for camera calibration.

**Open-Vocabulary 3DGS with Per-Scene Optimization.** LangSplat (Qin et al., 2024) pioneers open-vocabulary 3D Gaussian Splatting (3DGS) by distilling knowledge from vision-language models such as CLIP (Radford et al., 2021) and DINO (Caron et al., 2021). Similarly, Fmgs (Zuo et al., 2025) and Feature 3DGS (Zhou et al., 2024) explicitly inject vision-language features into the 3DGS pipeline. Building upon these approaches, LEGaussians (Shi et al., 2024) and GOI (Qu et al., 2024) introduce quantization techniques to compress high-dimensional semantic embeddings into compact Gaussian representations. Ji et al. (2025) proposes a Feature Grid Mapping strategy to accelerate open-vocabulary queries for high-resolution reconstruction. Methods such as OpenGaussian (Wu et al., 2024) and Gaussian Grouping (Ye et al., 2024) leverage 2D open-vocabulary segmentation tools like SAM (Kirillov et al., 2023) to assign semantic labels to rendered images, without explicitly embedding semantics into the Gaussians themselves.

**Generalizable Open-Vocabulary 3DGS.** Building on generalizable 3DGS, GSemSplat (Wang et al., 2024b) extends DUSt3R (Wang et al., 2024a) and Splatt3R (Smart et al., 2024) by incorporating semantic prediction heads to jointly estimate open-vocabulary features together with Gaussian parameters, thereby enabling feed-forward scene reconstruction. Similarly, SparseLGS (Hu et al., 2024) leverages MASt3R (Leroy et al., 2024) and introduces a multi-view semantic alignment strategy to achieve generalizable Gaussian semantic reconstruction from sparse input images. LSM (Fan et al., 2024b) further explores semantic anisotropic Gaussians for reconstructing explicit radiance fields from only two unposed images, supporting real-time 3D perception and scene understanding. More recent works, such as GaussTR (Jiang et al., 2025) and Uni3R (Sun et al., 2025), advance open-vocabulary 3DGS by achieving higher-quality occupancy prediction and scene understanding. Nevertheless, a major limitation remains: these open-vocabulary generalizable methods often struggle to reconstruct regions outside the narrow visual field covered by the sparse input views.

## 3 APPROACH

As illustrated in Figure 2, OGGSplat comprises three main stages. First, in Section 3.1, we initialize an open-vocabulary 3D Gaussian representations from the unposed image pairs. Next, Section 3.2 introduces the RGB-semantic consistent inpaintor, where we propose a bidirectional control mechanism to ensure pixel-level alignment between inpainted semantic maps and RGB images. The semantic map guides the image completion process, while the inpainted image, in turn, refines the semantic features. Finally, to allow the semantic-aware 3D Gaussian structure to grow consistently with the generated content, we design a progressive open-vocabulary Gaussian growing strategy, detailed in Section 3.3. The second and third stages are applied iteratively to gradually expand the Gaussian representation beyond the initial field-of-view. In practical usage, OGGSplat takes as input any two uncalibrated images and processes them through the above three stages to produce an expanded open-vocabulary 3D Gaussian scene. This enables real-time rendering of both RGB images and their corresponding semantic feature maps from wider arbitrary viewpoints.

### 3.1 GENERALIZABLE OPEN-VOCABULARY GAUSSIAN INITIALIZATION

**Gaussian Reconstruction.** Given any two uncalibrated but overlapping images $I_1, I_2 \in \mathbb{R}^{H \times W \times 3}$ with height $H$ and width $W$, we adopt Splatt3R (Smart et al., 2024) to reconstruct an initial Gaussian $\mathcal{G}_0 \in \mathbb{R}^{N \times d}$ via a shared backbone, cross-attention interactions and multiple Gaussian heads. The number of Gaussian primitives $N = 2 \times H \times W$ corresponds to the total number of image pixels, while each Gaussian feature of dimension $d$ is composed of the following components: (1) a 3D point position $p \in \mathbb{R}^3$, (2) a position offset $p_\Delta \in \mathbb{R}^3$, defining the Gaussian center $\mu = p + p_\Delta$, (3) a rotation quaternion $q \in \mathbb{R}^4$ and a scale vector $s \in \mathbb{R}^3$, together determining the covariance matrix $\Sigma$,

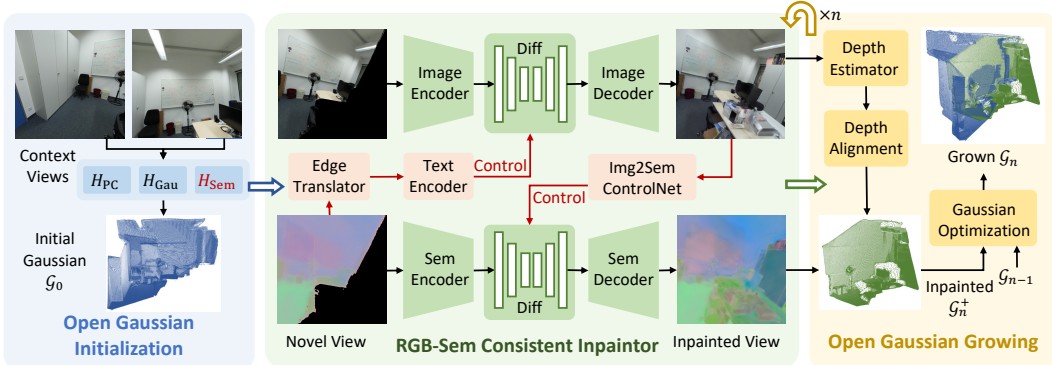

Figure 2: **OGGSplat Architecture.** We first initialize an open-vocabulary Gaussian reconstruction, injecting semantics via an additional semantic head. Then, the RGB-semantic consistent inpaintor applies bidirectional controls between images and semantic maps to ensure semantic plausibility and spatial alignment. Finally, the inpainted regions are lifted back to 3D and optimized to expand the Gaussians. The last two stages are performed iteratively to progressively grow the Gaussians.

(4) an opacity scalar $\alpha \in \mathbb{R}$, controlling the transparency of the Gaussian, and (5) a view-dependent appearance embedding represented by spherical harmonics $\mathbf{S} \in \mathbb{R}^{3 \times d_{\text{color}}}$ of $d_{\text{color}}$ degrees.

**Open Feature Injection.** To incorporate open-vocabulary clues, we introduce an additional semantic head $H_{\text{sem}}$ to predict semantic parameters $f \in \mathbb{R}^{d_{\text{sem}}}$ for each Gaussian primitive, inspired by GSemSplat (Wang et al., 2024b). Following common practice (Ye et al., 2024; Wang et al., 2024b), we set the semantic embedding dimension $d_{\text{sem}} = 16$ to reduce the computational overhead during Gaussian rendering. To supervise the predicted semantic features $f$, we adopt the well-optimized vision-language APE (Shen et al., 2024) model to efficiently obtain pixel-dense open-vocabulary semantic supervision signals $F^{\text{gt}} \in \mathbb{R}^{H \times W \times d_{\text{APE}}}$, where the APE semantic feature dimension $d_{\text{APE}} \gg d_{\text{sem}}$. To align the dimensionality, we train an autoencoder composed of a down-projection encoder $\mathcal{E}_{\downarrow}$ that maps the APE features to $d_{\text{sem}}$, and a corresponding decoder $\mathcal{D}_{\uparrow}$ that reconstructs the original features with minimal information loss. The semantic learning objective in this stage is formulated as a cosine similarity loss:

$$\mathcal{L}_{\text{sem}} = \sum_{v} \sum_{h,w} \left( 1 - \cos \left( f_{v,h,w}, \mathcal{E}_{\downarrow} \left( f^{gt}_{v,h,w} \right) \right) \right), \tag{1}$$

where $h \in [0, H), w \in [0, W)$ denote pixel coordinates and $v$ represents target view index. The semantic feature $f_{v,h,w}$ is computed with $\alpha$-blending, analogous to that used for RGB rendering.

## 3.2 RGB-SEMANTIC CONSISTENT INPAINTOR

Once the initial Gaussian $\mathcal{G}_0$ is reconstructed, we render RGB images $I_v$ and their corresponding semantic maps $F_v$ from novel viewpoints $v$. However, when rendering outside the vision cone of the context views, hollow regions often appear due to out-of-view areas and occlusion variations, as illustrated in Figure 1. While pre-trained inpainting diffusion models (Rombach et al., 2022; Lugmayr et al., 2022; Xie et al., 2023) can partially address this issue, maintaining pixel-wise consistency between inpainted images and their semantic maps remains challenging. This spatial misalignment will be inherited by the following Gaussian growing process and can lead to increasingly severe semantic inconsistencies as the scene expands. Fortunately, we observe that although the semantic modality introduces challenges, it also offers valuable guidance: the partial semantic information, especially around the boundaries of incomplete regions, can be translated into explicit textual prompts to guide image inpainting. Symmetrically, inpainted RGB images can provide pixel-wise appearance cues to control semantic map completion. Therefore, we propose bidirectional controls between the RGB branch $\text{Diff}_{\text{rgb}}$ and the semantic branch $\text{Diff}_{\text{sem}}$, allowing them to mutually enhance each other during the inpainting process.

**Semantic-to-RGB Control.** To define the inpainting mask that determines whether a pixel should be inpainted, we rely on the rendered opacity $\alpha$ of each pixel. Similar to color rendering, we render an opacity map $A$, and then derive the inpainting mask $M_v$ for each view $v$ by applying a pre-defined

threshold $\tau$. For simplicity, we omit the view subscript $v$ in the following discussion.

$$A_{h,w} = \sum_{i \in \Theta_{h,w}} \alpha_i \prod_{j=1}^{i-1}(1 - \alpha_j), \qquad M_{h,w} = \mathbb{1}\left[A_{h,w} < \tau\right], \tag{2}$$

where $\Theta_{h,w}$ denotes the set of Gaussians contributing to the pixel at coordinate $(h, w)$.

Then we design an *Edge Translator* to extract semantic concepts near the inpainting boundaries defined by the mask $M$, providing clearer guidance for hollow region inpainting. Specifically, we first identify pixels along the boundary as $\Omega_{\text{edge}}$. The corresponding semantic features $f_{\text{edge}}$ of these boundary pixels are then decoded into a higher-dimensional space using our pre-trained decoder $\mathcal{D}_\uparrow$:

$$g_{\text{edge}} = \mathcal{D}_\uparrow(f_{\text{edge}}), \text{ for pixels in } \Omega_{\text{edge}} \tag{3}$$

Simultaneously, we prepare a set of candidate classes $\mathcal{C}_{\text{cand}}$, consisting of the top 100 semantic categories in our training dataset. These categories are encoded into the same feature space as $g_{\text{edge}}$. We then compute the cosine similarity between $g_{\text{edge}}$ and $g_{\text{cand}}$ to perform pixel-wise segmentation:

$$c_{\text{edge}} = \text{argmax}_{c_i \in \mathcal{C}_{\text{cand}}} \cos(g_{\text{edge}}, g_{c_i}), \tag{4}$$

In this way, we can obtain a set of semantic categories $\mathcal{C}_{\text{edge}}$ that are most relevant to the inpainting region. Based on these categories, we generate a prompt text $T$ in the format of *"a room with* $\text{cate}_1$, $\text{cate}_2$, *..., and* $\text{cate}_i$*"*, which is used to guide the diffusion-based RGB image inpainting model:

$$I^{\text{inp}} = \text{Diff}_{\text{rgb}}(I, M, T), \tag{5}$$

**RGB-to-Semantic Control.** Inspired by ControlNet (Zhang et al., 2023), we also design an RGB-to-Semantic control module to ensure that the generated semantic content aligns well with the corresponding regions in the RGB image. Formally, the completed semantic map is computed as:

$$F^{\text{inp}} = \text{Diff}_{\text{sem}}(F, M, T, \text{ControlNet}(I^{\text{inp}})), \tag{6}$$

where $F$ is the incomplete rendered semantic feature map, and $\text{ControlNet}(I^{\text{inp}})$ denotes the control module conditioned on the inpainted image $I^{\text{inp}}$. Please refer to the ControlNet paper or Appendix B.4 for further details. This module guides the semantic generation process, ensuring both structural and appearance consistency between the predicted semantic features and the RGB content.

### 3.3 OPEN-VOCABULARY GAUSSIAN GROWING

Obtaining the inpainted RGB images and semantic feature maps from selected views is not the final step of our pipeline. These results must be aggregated back into the initial Gaussian $\mathcal{G}_0$ to enable real-time rendering from arbitrary novel viewpoints. For selected anchor views $V = \{v_3, v_4, \cdots, v_a\}$, we perform iterative inpainting and progressively incorporate the newly completed regions into the Gaussian. At each iteration $n$, a new view is rendered based on the currently aggregated Gaussians $\mathcal{G}_{n-1}$ and the newly inpainted content $\mathcal{G}_n^+$ is fused into this representation. Below, we break down a single iteration and describe the Gaussian growing process in detail.

The inpainted image $I^{\text{inp}}$ and semantic map $F^{\text{inp}}$ will serve as supervision targets for the newly grown Gaussians. However, establishing 3D geometry from a single novel view is inherently ill-posed, especially in regions that are newly generated during inpainting. To enrich these views with structural knowledge, we adopt custom depth estimation model (Piccinelli et al., 2024; Yang et al., 2024a;b) to predict an absolute depth map $D^{\text{inp}}$ from $I^{\text{inp}}$. This depth map is then used to lift pixels back into 3D space, forming a point cloud in the global coordinate system. The resulting 3D points are used to initialize the position of the incremental Gaussian set $\mathcal{G}^+$, which is progressively integrated into the scene representation.

$$P^+ = \text{proj}(D^{\text{inp}}, v^{\text{inp}}, v_1, K) \cdot \beta, \text{ where } \beta = \frac{\sqrt{\frac{1}{M}\sum_{i=1}^{M}\left\|p_i^{\text{ori}}\right\|_2^2}}{\sqrt{\frac{1}{N}\sum_{i=1}^{N}\left\|p_i^{\text{new}}\right\|_2^2}} \tag{7}$$

where $v^{\text{inp}}$ and $v_1$ are the camera poses corresponding to the images $I^{\text{inp}}$ and $I_1$, respectively, and $K$ denotes the intrinsic camera parameters. The scale factor $\beta$ is introduced to align the newly projected

point cloud with the original 3D space in terms of depth. $p^{\mathrm{ori}}, p^{\mathrm{new}}$ denote the original and newly projected 3D points within the overlapping regions, while $M$ and $N$ represent the respective number of points in each set. It is worth noticing that scaling point coordinates alone does not ensure perfect alignment. Nonetheless, it offers an efficient and approximate initialization, since the entire scene is constructed with respect to the normalized coordinate system of the first view.

At the $n^{th}$ iteration, after merging $\mathcal{G}_{n-1}$ with the newly initialized Gaussians $\mathcal{G}_n^+$, we perform efficient per-scene optimization to update the grown Gaussian $\mathcal{G}_n$. This optimization is supervised by the original sparse context views, previously and newly inpainted views. The objective function is:

$$\mathcal{L} = \lambda_{\mathrm{rgb}} \cdot \mathcal{L}_{\mathrm{rgb}} + \lambda_{\mathrm{feat}} \cdot \mathcal{L}_{\mathrm{feat}}, \tag{8}$$

$$\text{where } \mathcal{L}_{\mathrm{rgb}} = \lambda_1 \cdot \mathcal{L}_{\mathrm{L1}}(I^{\mathrm{r}}, I^{\mathrm{inp}}) + \lambda_2 \cdot \mathcal{L}_{\mathrm{SSIM}}(I^{\mathrm{r}}, I^{\mathrm{inp}}), \text{ and } \mathcal{L}_{\mathrm{feat}} = 1 - \cos(F^{\mathrm{r}}, F^{\mathrm{inp}}) \tag{9}$$

where $\lambda_1$ and $\lambda_2$ balance pixel-wise accuracy and perceptual similarity, while $\lambda_{\mathrm{rgb}}$ and $\lambda_{\mathrm{feat}}$ control the overall contributions of the photometric and semantic losses, respectively. $I^{\mathrm{r}}, F^{\mathrm{r}}$ denote the rendered RGB images and semantic features from the optimizing Gaussian from $v^{\mathrm{inp}}$.

## 4 EXPERIMENTS

### 4.1 THE OPEN-VOCABULARY GAUSSIAN OUTPAINTING (OVGO) BENCHMARK

To effectively evaluate both the visual fidelity and semantic plausibility of OGGSplat in extrapolated regions, we introduce a novel Open-Vocabulary Gaussian Outpainting benchmark based on the ScanNet++ (Yeshwanth et al., 2023) dataset. Detailed information can be found in the Appendix B.1.

**Data Composition.** The OVGO benchmark covers all 50 validation scenes from ScanNet++. For each scene, we select *1 image pair* as the context views of inputs. To ensure consistency in data sampling and maintain temporal coherence, the context views are chosen as the $1^{\mathrm{st}}$ and $10^{\mathrm{th}}$ frames. This selection introduces moderate viewpoint variation while preserving semantic continuity, enabling a more meaningful evaluation of extrapolated content. For evaluation, we uniformly sample *16 novel camera poses* within a horizontal range of $[-60°, 60°]$ and a vertical range of $[-20°, 20°]$ around the pose of the context image $I_1$. Novel RGB images and semantic maps are directly rendered from the reconstructed Gaussians at these poses and used as evaluation samples. Considering randomness, we repeat the experiment five times and report the average results.

**Visual Fidelity Evaluation.** We adopt the Fréchet Inception Distance (FID) (Heusel et al., 2017) to evaluate the statistical similarity between rendered and real images. For FID computation, all images from the validation split of the ScanNet++ dataset are used as the reference distribution. FID is then calculated between this reference distribution and the distribution of the newly rendered images. However, we observe that the limited number of generated images can negatively affect the stability of the FID metric. To address this, we increase the context views from *one pair* to *ten pairs* per scene, while maintaining a frame interval of 10 within each pair. This expands the number of newly rendered images by a factor of ten, resulting in a more stable and reliable FID evaluation.

**Semantic Plausibility Evaluation.** We assess semantic plausibility by performing open-vocabulary semantic segmentation on novel views. To enable a more comprehensive analysis, we separately evaluate performance in low-confidence ($\mathrm{mIoU}_L$) and high-confidence ($\mathrm{mIoU}_H$) regions. Low-confidence regions are defined as pixels in novel views where the accumulated opacity of the initial Gaussians falls below 0.3, corresponding to occluded or out-of-view areas. Since evaluating these regions emphasizes semantic consistency in extrapolated areas, $\mathrm{mIoU}_L$ serves as our primary metric. For reference, we additionally report $\mathrm{mIoU}_H$, which measures semantic plausibility in high-confidence regions, defined as pixels where the accumulated opacity of the initial Gaussians exceeds 0.3, thus indicating areas reliably rendered by the original representation.

Since ground truth semantic annotations are unavailable for extrapolated regions, we generate them using five state-of-the-art open-vocabulary 2D semantic segmentation models (Xu et al., 2023; Shen et al., 2024; Zeng et al., 2024; Yu et al., 2023; Jiao et al., 2024). Their predictions are aggregated via a majority voting scheme. To assess the quality of semantic segmentation, we follow the protocol in (Kerr et al., 2023; Shi et al., 2024; Qin et al., 2024) by computing a relevancy score for each text query. More details about relevancy score are provided in Appendix B.5. To ensure generality, we retain only those predicted mask regions with a relevancy score exceeding 50% as the final binary

Table 1: **Open-Vocabulary Gaussian Outpainting (OVGO) benchmark results.** We compare generative metric FID and semantic metric mIoU (%) between OGGSplat and previous methods.

| Methods | Generation | Segmentation (IoU↑) | | | | | | | | | | | |
| | FID↓ | $mIoU_H$ | $mIoU_L$ | wall | ceiling | floor | table | door | (s)cabinet | chair | (b)shelf | box | bed |
|---|---|---|---|---|---|---|---|---|---|---|---|---|---|
| LangSplat | 50.4 | 13.5 | 6.9 | 29.0 | **13.4** | 15.8 | 1.8 | 4.0 | 1.3 | 2.5 | 0.0 | 0.8 | 0.0 |
| Splatt3R | 46.4 | 24.9 | 6.0 | 10.1 | 2.1 | 18.9 | 5.1 | 0.0 | 1.6 | 13.8 | 0.3 | 0.0 | 2.3 |
| **OGGSplat** | **37.5** | **25.4** | **17.6** | **45.6** | 0.1 | **58.3** | **13.3** | **5.4** | **3.7** | **21.4** | **7.4** | **3.1** | **18.0** |

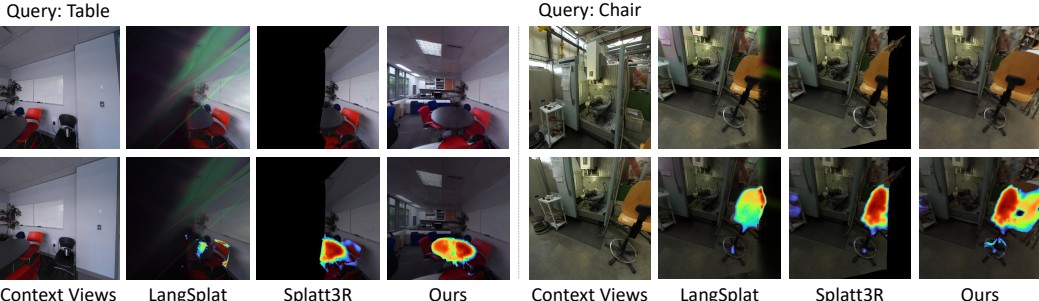

Figure 3: **Qualitative comparisons between LangSplat, Splatt3R, and OGGSplat on the OVGO benchmark.** The first row presents RGB images rendered from novel, out-of-scope viewpoints. The second row visualizes the heatmap when querying different text concepts.

mask. This filtering mechanism makes our evaluation suitable even for images where a specific category may be absent. During evaluation, we focus on 10 commonly used semantic categories selected from the top 20 classes in ScanNet++, such as *wall*, *floor*, *chair*, *table*, and others.

## 4.2 MAIN RESULTS

**Baseline Methods for Comparison.** We select two representative open-vocabulary Gaussian baselines for comparison: LangSplat (Qin et al., 2024), a per-scene optimization model, and Splatt3R (Smart et al., 2024), a generalizable model. LangSplat relies heavily on accurate initialization via COLMAP (Schonberger & Frahm, 2016), which becomes unreliable when only two input images are available. To address this limitation and enable fair comparison, we initialize LangSplat using point cloud positions predicted by Splatt3R, allowing the model to focus more effectively on learning semantic representations. Meanwhile, as vanilla Splatt3R does not support open-vocabulary semantic prediction in its original form, we extend it with a semantic head trained in our first stage in Section 3.1. During evaluation, for all models, we consider only the regions rendered by Gaussians with an accumulated opacity greater than 0.01 as valid predictions for computing the IoU scores. This threshold filters out low-confidence regions and ensures consistency across models.

**Quantitative Comparisons.** In Table 1, we compare LangSplat (Qin et al., 2024), Splatt3R (Smart et al., 2024), and OGGSplat on the OVGO benchmark. OGGSplat consistently outperforms the baselines by a significant margin on both visual fidelity (FID) and semantic plausibility (mIoU). It's worth noticing that the overall FID remains relatively high across all methods. The main reason is the limited number of context pairs available in the validation set, which constrains data diversity. We are unable to sample more pairs because some scenes in the ScanNet++ validation set are relatively small. To maintain a consistent sampling ratio across all validation scenes, we limit the number of context pairs to 10 per scene. Regarding semantic plausibility, in the low-confidence regions, which are our primary focus, OGGSplat achieves notably better performance on common large objects such as *chair*, *table*, and *bed*. However, the model performs relatively worse on the *ceiling* class. We attribute this to the limitations of the APE encoding, as well as the difficulty of the Splatt3R backbone in distinguishing between the *ceiling* and *wall* with similar appearance in color and texture. We believe this limitation can be addressed in future work by leveraging more powerful vision-language models and more superior generalizable Gaussian reconstruction methods. For high-confidence regions, although these are not our primary focus, it is still notable that OGGSplat slightly outperforms the Splatt3R backbone. This improvement is mainly due to our method's ability to recover small chunks within the view cone that initially have relatively low confidence.

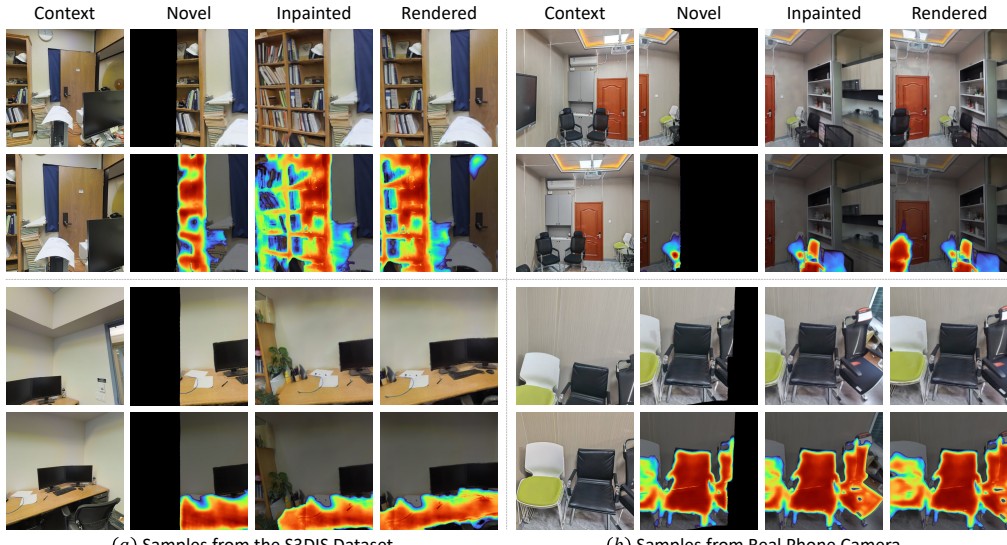

(a) Samples from the S3DIS Dataset      (b) Samples from Real Phone Camera

Figure 4: **Zero-shot generalization to out-of-distribution data.** (a) Context views are taken from the S3DIS dataset, with queries *bookshelf* and *table*. (b) Context views are captured using a **phone camera**, with the query *chair*. In both cases, we directly apply the model trained on ScanNet++.

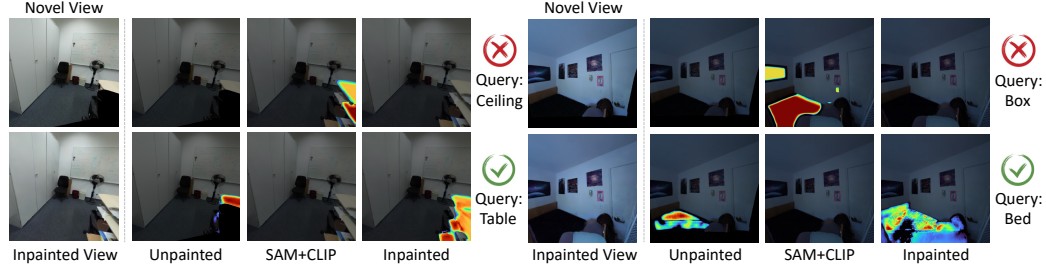

Figure 5: **Ablations on the effect of semantic diffusion model.** We compare open-vocabulary predictions between the SAM+CLIP offline method and our semantic diffusion inpainting module.

**Qualitative Comparisons.** We conduct extensive qualitative comparisons with baseline methods and illustrate them in Figure 3. OGGSplat performs better in both novel rendered images and open-vocabulary querying. Regarding rendered images, LangSplat tends to overfit the context views, resulting in blurry renderings from novel viewpoints, even when the Gaussian positions have been initialized. Splatt3R, on the other hand, exhibits large black regions in areas outside the input views. In contrast, OGGSplat reasonably extrapolates unseen regions by leveraging semantic information. Regarding open-vocabulary querying, both LangSplat and Splatt3R are limited to input vision cones. OGGSplat, however, is capable of accurately identifying and querying objects even in previously unseen regions, demonstrating stronger generalization and semantic understanding capabilities.

**Zero-Shot Generalization to Out-of-Distribution Data.** Apart from ScanNet++ used for training, we also test OGGSplat's zero-shot ability on different distributions. As shown in Figure 4, OGGSplat successfully reconstructs expanded semantic-aware scenes on the S3DIS (Armeni et al., 2016) dataset. We further demonstrate the practicality of OGGSplat on portable devices in column (b), where the inputs are captured by a phone camera. The inpainted image and category-specific query show promising results, highlighting OGGSplat's potential for applications in daily life.

### 4.3 ABLATION STUDIES

In Section 3.2, we introduced the RGB-semantic consistent inpainting module. In this section, we first highlight the importance of the semantic diffusion branch, followed by comprehensive ablations on the OVGO benchmark to evaluate the effectiveness of the proposed bidirectional control strategy.

**Semantic Diffusion Model.** To obtain reliable semantics for inpainted regions, we train a semantic diffusion module. A simple alternative is using an offline open-vocabulary semantic segmentation

Table 2: **Ablations on the OVGO benchmark evaluating the impact of the bidirectional control strategy.** The performance is measured by mIoU (%) across various semantic categories.

| Control Type | | Segmentation Results (IoU ↑) | | | | | | | | | | |
|---|---|---|---|---|---|---|---|---|---|---|---|---|
| S→RGB | RGB→S | $mIoU_L$ | wall | ceiling | floor | table | door | (s)cabinet | chair | (b)shelf | box | bed |
| ✗ | ✓ | 16.6 | **45.8** | 0.1 | 56.8 | 12.3 | 4.6 | 2.8 | 19.3 | 6.3 | **3.6** | 15.1 |
| ✓ | ✗ | 14.4 | 43.0 | 0.1 | 47.6 | 10.3 | 5.0 | 3.5 | 16.6 | 2.4 | 2.5 | 12.7 |
| ✓ | ✓ | **17.6** | 45.6 | 0.1 | **58.3** | **13.3** | **5.4** | **3.7** | **21.4** | **7.4** | 3.1 | **18.0** |

Figure 6: **Qualitative comparison of bidirectional control.** Row 1 shows the context images and the incomplete renderings from novel views. Rows 2 to 4 correspond to the ablation settings in Table 2, where each variant removes one of the control mechanisms to examine its individual effect.

model, e.g., SAM (Kirillov et al., 2023)+CLIP (Radford et al., 2021). However, this often causes semantic inconsistency with the original Gaussian, especially for partially visible objects (Figure 5). In contrast, our semantic diffusion model preserves semantic consistency in unpainted areas and leverages context priors to improve accuracy in the inpainted regions, ensuring new content aligns with existing scene semantics and enhancing overall reconstruction quality.

**Semantic-to-RGB Control.** With access to open-vocabulary semantics, we propose an edge translator to extract semantic cues from the Gaussian boundaries and guide the completion. In the first row of Table 2, we remove the edge translator and instead use a generic description ("a room") as the text prompt. As a result, semantic segmentation performance across most categories decreases. This degradation is also evident in the qualitative comparison in Figure 6, where the generated content appears more ambiguous and less semantically grounded. These results validate the effectiveness of our semantic-to-RGB control in guiding high-fidelity, semantically consistent Gaussian growth.

**RGB-to-Semantic Control.** In OGGSplat, the semantic inpainting model is explicitly controlled by inpainted images. We remove it in the second row of Table 2 and the third row of Figure 6. Without RGB-to-semantic control, the generated RGB images and semantic maps exhibit poor spatial alignment, leading to significantly degraded segmentation accuracy. In contrast, introducing the RGB-to-semantic control clearly improves spatial consistency and yields much better performance.

## 5 CONCLUSION

In this paper, we present OGGSplat, a generalizable open-vocabulary 3D Gaussian growing method for expanded field-of-view. By leveraging semantic cues and introducing RGB-semantic consistent inpainting with bidirectional control, OGGSplat effectively expands the view while maintaining visual fidelity and semantic coherence. Out-of-view regions are then progressively refined through efficient Gaussian optimization. We also propose the Open-Vocabulary Gaussian Outpainting benchmark for semantic-aware evaluation on expanded regions. Extensive experiments show that OGGSplat effectively extrapolates beyond the input view cone while keeping RGB-semantic alignment, marking a significant step forward in generalizable and flexible open-vocabulary 3D reconstruction.

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

## A ADDITIONAL EXPERIMENTAL RESULTS

### A.1 VIDEO RESULTS

To provide a more comprehensive and intuitive visualization of our method, we include video results in the supplementary ZIP file. Specifically, we present visualizations across five different scenes. For each scene, we showcase the rendering results of both Splatt3R (Smart et al., 2024) and OGGSplat under continuous camera views. Additionally, we provide the corresponding relevance score heatmaps under a specific open-vocabulary query, enabling a direct comparison of semantic understanding across the two methods. As clearly demonstrated, our model effectively extrapolates to unseen regions while maintaining both high visual fidelity and semantic plausibility.

### A.2 MORE ANALYSIS ON VISUAL FIDELITY

In Table 1, we use FID to evaluate the consistency and diversity between the distributions of rendered novel views and the original dataset. Here, we supplement this evaluation with additional metrics, including CLIP score, DINO score, and LLM score, providing a more comprehensive assessment of the rendered image quality.

**CLIP Score.** We use the CLIP (Radford et al., 2021) score to measure the semantic similarity between the novel (outpainted) regions and the original contextual regions. For each sample in the ScanNet++ validation set, we randomly generate masks and use MaskCLIP (Ding et al., 2022) to extract features from both the masked and unmasked regions. The cosine similarity between these features is computed and averaged across all samples to obtain the oracle score, serving as an upper bound for contextual alignment. We further compute CLIP scores for our method as well as LangSplat and Splatt3R, comparing the novel regions against the original context regions.

**DINO Score.** While the CLIP score emphasizes semantic coherence at the boundary between expanded and original content, it is less sensitive to style differences. To address this, we evaluate edge regions using the DINO score (Caron et al., 2021). DINO, as a well-established visual representation model, captures both semantic and texture features, including style, providing a more holistic assessment of visual fidelity. The measurement procedure for all models and the oracle is kept fully consistent with that of the CLIP score, ensuring a fair and comparable evaluation across metrics.

**LLM Score.** To provide a more global assessment, we employ the well-trained vision-language model Qwen2.5-VL-7B-Instruct (Bai et al., 2025) as an expert evaluator. The model is prompted to score inpainted images based on authenticity, style continuity, and structural integrity. Specifically, we use the following prompt: "Please assess the image's authenticity, style continuity, and structural integrity, and assign a score between 1 and 100. Output your result strictly in the format: The score is XXX." This approach allows for a holistic evaluation that complements the CLIP and DINO scores, capturing both visual fidelity and stylistic coherence.

Table 3 presents the results for all three metrics. We observe that our method consistently approaches the oracle scores across all measures, demonstrating superior performance in semantic coherence, style continuity, and structural integrity.

Table 3: Comparison of CLIP score, DINO score, and LLM score across different models.

| Model | CLIP Score | DINO Score | LLM Score |
|---|---|---|---|
| Oracle | 53.7 | 84.9 | 76.5 |
| LangSplat | 47.9 | 82.3 | 66.9 |
| Splat3R | 44.6 | 74.7 | 71.8 |
| OGGSplat | 50.8 | 83.4 | 73.6 |

### A.3 ABLATION ON SEPARATE DIFFUSION UNET

To enable the generation of both spatially consistent RGB images and semantic content, we train two separate diffusion models: $\text{Diff}_{\text{rgb}}$ and $\text{Diff}_{\text{sem}}$, and enforce spatial consistency between them using a ControlNet (Zhang et al., 2023) based approach. A simpler alternative would be to employ a single

Table 4: Ablation on semantic type for inpainting.

| Methods | $mIoU_L$ | wall | ceiling | floor | table | door | cabinet | chair | shelf | box | bed |
|---------|----------|------|---------|-------|-------|------|---------|-------|-------|-----|-----|
| Implicit Condition | 17.0 | 44.7 | 0.1 | 58.8 | 13.5 | 3.5 | 2.7 | 22.2 | 5.8 | 3.6 | 14.8 |
| Explicit Condition | 17.6 | 45.6 | 0.1 | 58.3 | 13.3 | 5.4 | 3.7 | 21.4 | 7.4 | 3.1 | 18.0 |

Table 5: Ablation study of the depth alignment module evaluated by Chamfer Distance (CD).

| Alignment Strategy | CD1↓ | CD2↓ |
|--------------------|------|------|
| No Depth Alignment | 0.28 | 0.48 |
| Bounding Box Alignment | 0.21 | 0.32 |
| EMS Alignment (Ours) | **0.13** | **0.17** |

Table 6: Comparison of time consumption.

| Stage | LangSplat | OGGSplat |
|-------|-----------|----------|
| Gaussian Init. | 0.6s | 0.6s |
| Inpainting | - | 5.8s |
| Gaussian Opt. | 157s | 27.5s |
| Total | 157.6s | 33.9s |

shared diffusion UNet based on an image diffusion model (Rombach et al., 2022), modified to allow additional semantic inputs and outputs by adjusting the input and output convolutional channels. However, our experiments show that this approach fails to produce meaningful RGB and semantic outputs. As illustrated in Fig. 7, using a hybrid (shared) diffusion UNet leads to severe distortions in both RGB images and semantic content. We think that this failure is due to the significant differences between the latent spaces of the RGB image VAE and the semantic VAE, which makes it difficult for a single UNet to learn consistent mappings in both domains. These results highlight the effectiveness and necessity of our separate $Diff_{sem}$ model and the corresponding control module design.

### A.4 Ablation on Semantic Type for Inpainting

We utilize explicit text representations as conditions rather than edge features since our inpainting model is fine-tuned from a pre-trained image diffusion model, which is originally conditioned on text prompts via the CLIP text encoder. Maintaining the same conditioning modality enables us to better leverage the prior knowledge acquired during pretraining, leading to more controllable and semantically accurate inpainting. In contrast, using implicit edge features as conditioning can result in less effective control due to potential embedding space misalignment, particularly given the relatively small scale of our fine-tuning dataset compared to the pretraining corpus.

Moreover, explicit text conditioning provides an additional advantage over implicit feature-based conditioning: user-specified descriptions can be directly appended to the original text prompt, allowing for more flexible and intuitive control over the generated content.

To validate this design choice, we perform a quantitative ablation study. In the first row of Table 4, we replace the explicit text prompt with implicit APE features as the inpainting condition. The results demonstrate that explicit text prompts consistently achieve higher segmentation mIoU, with notable improvements observed for categories such as *cabinet*, *shelf*, and *bed*.

### A.5 Ablation on Depth Alignment Module

The design of the depth alignment module is a crucial component for minimizing the distance between newly added Gaussian points and the original Gaussian points. To demonstrate its effectiveness, we conduct an ablation study. Specifically, we employ Chamfer Distance L1 (CD1) and Chamfer Distance L2 (CD2) to quantify the spatial discrepancy between the reprojected Gaussian points and the original points within overlapping regions across multiple views. The experimental results are presented in Table 5, where the rows from top to bottom correspond to: no depth alignment, bounding box diagonal ratio-based alignment, and our proposed EMS ratio-based alignment.

The results highlight the importance of the depth alignment module. Introducing the alignment method significantly reduces CD values compared to the case without alignment. Notably, our EMS-based alignment consistently outperforms the bounding box-based strategy. We attribute this improvement to the robustness of EMS alignment against outliers, which can otherwise negatively impact the accuracy of alignment estimation.

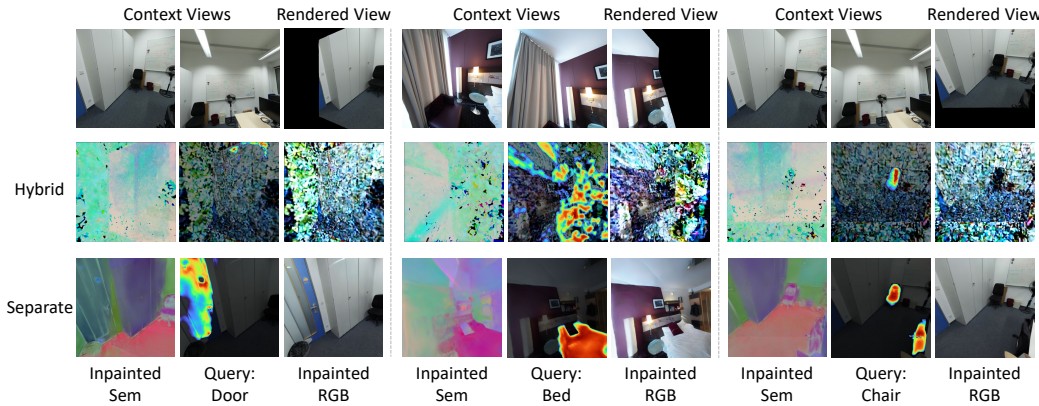

Figure 7: Qualitative comparison between hybrid (shared-weight) and separate diffusion UNet architectures. Row 1 shows the context images along with the incomplete renderings from novel views. Row 2 presents the results by using a hybrid UNet that jointly predicts RGB image and semantic content using shared weights. Row 3 shows the results from our proposed architecture with two separate UNets: one for RGB image synthesis and the other for semantic prediction.

### A.6 DISCUSSION ON TIME CONSUMPTION

We provide a detailed comparison of the runtime between our method and LangSplat for a single user-defined novel view in Table 6. Although our pipeline introduces an additional inpainting stage, which is absent in LangSplat, this stage significantly accelerates the subsequent Gaussian optimization process. By leveraging the inpainted content together with UniDepth, we achieve more effective Gaussian initialization and optimization. Specifically, while the inpainting step incurs an extra cost of 5.8s, the overall Gaussian optimization time is drastically reduced (27.5s vs. 157s in LangSplat), leading to a substantial reduction in the total runtime (33.9s vs. 157.6s).

Furthermore, our pipeline naturally supports an arbitrary number of user-defined novel views, with the computational cost scaling linearly with the number of views, offering controllability and flexibility. After optimization with the inpainted view(s), our method further enables real-time rendering for additional novel views, consistent with the runtime characteristics of existing Gaussian Splatting-based approaches.

## B IMPLEMENTATION DETAILS

### B.1 SCANNET++ DATASET

ScanNet++ dataset (Yeshwanth et al., 2023) provides high-quality 3D geometry along with high-resolution RGB images of various indoor environments. Following the protocol introduced by Splatt3R, originally designed for 3D reconstruction, we adopt the standard training split comprising 230 scenes and the validation split containing 50 scenes. Following (Smart et al., 2024; Wang et al., 2024b), we also discard frames missing reliable depth information. All selected frames are uniformly cropped and resized to a spatial resolution of $512 \times 512$.

### B.2 TRAINING SETTINGS

To provide a clearer overview of the experimental configurations used at different training stages, we summarize the details in Table 8. The table includes the settings for all key components that need to be trained in our method, namely the generalizable open-vocabulary Gaussian initialization module, RGB UNet, semantic VAE (Kingma et al., 2013), semantic UNet, ControlNet (Zhang et al., 2023), and the open-vocabulary Gaussian growing process.

**Generalizable Open-Vocabulary Gaussian Initialization**. We adopt the pretrained Splatt3R model and freeze its backbone, which is responsible for predicting the basic Gaussian attributes. We then train only the newly added semantic head, denoted as $H_{\text{sem}}$. During training, we use two context images as input and supervise the model by rendering three target views from the training

Table 8: Experiment settings for different training stages.

| Config | Gaussian Init. | RGB-Semantic Consistent Inpaintor | | | | Gaussian Growing |
| | | RGB UNet | Sem. VAE | Sem. UNet | ControlNet | |
|---|---|---|---|---|---|---|
| optimizer | Adam | AdamW8bit | AdamW | AdamW8bit | AdamW8bit | Adam |
| learning rate | 1e-5 | 1e-5 | 6e-6 | 1e-5 | 1e-5 | hybrid (Table 7) |
| weight decay | 5e-2 | 1e-2 | 1e-2 | 1e-2 | 1e-2 | 0 |
| scheduler | multi-step | constant | cosine | constant | constant | exponential |
| batch size | 12 | 4 | 2 | 4 | 4 | 4 |
| accumulation steps | 1 | 2 | 4 | 2 | 2 | 1 |
| training iterations | 500,000 | 50,000 | 45,000 | 20,000 | 10,000 | 600 |
| GPU device | 8 RTX 3090 | 8 RTX 3090 | 8 RTX 3090 | 8 RTX 3090 | 8 RTX 3090 | 1 RTX 3090 |
| image size | 512×512 | 512×512 | 512×512 | 512×512 | 512×512 | 512×512 |

split. Following the setup in Splatt3R (Smart et al., 2024), the context images are selected such that at least 30% of the pixels in the second image have direct correspondences in the first image. Similarly, target images are chosen such that at least 30% of their content is visible in at least one of the context images.

**RGB-Semantic Consistent Inpaintor**. For RGB image inpainting model $\text{Diff}_{\text{sem}}$, we fine-tune a stable diffusion inpainting model (Rombach et al., 2022) to better align the generated appearance with realistic indoor scenes. In addition to standard RGB inpainting, we propose a novel diffusion-based feature inpainting model, denoted as $\text{Diff}_{\text{sem}}$, which consists of both a Variational Autoencoder (Kingma et al., 2013) (VAE) and a UNet architecture. This model enables semantic-aware inpainting in the feature space while maintaining consistency with the RGB domain. To ensure spatial consistency between the RGB and semantic contents, we train an auxiliary RGB control module inspired by ControlNet (Zhang et al., 2023) that guides the inpainting process in the feature space.

**Open-Vocabulary Gaussian Growing**. We set the horizontal and vertical outpainting angles to lie within the ranges of $[-60°, 60°]$ and $[-20°, 20°]$, respectively. To simplify this stage, we decouple the horizontal and vertical rotations: when the horizontal angle is non-zero, the vertical angle is set to zero, and vice versa. For each optimization round, to improve efficiency, we use two inpainted images and their corresponding semantic maps under symmetrical camera poses to provide the supervision signal. Moreover, the selected camera view pairs are arranged to exhibit progressively increasing angular differences, thereby enabling a gradual and

Table 7: Learning rates for different Gaussian parameters.

| Parameter | Learning Rate |
|---|---|
| point position $\mu$ | 1e-2 |
| rotation quaternion $q$ | 1e-3 |
| scale vector $s$ | 5e-3 |
| opacity scalar $\alpha$ | 5e-2 |
| spherical harmonics $\mathbf{S}$ | 2.5e-2 |
| semantic feature $f$ | 2.5e-3 |

progressive Gaussian growing process. Specifically, denoting the camera rotation angles in the horizontal and vertical directions as $(\theta_h, \theta_v)$, the sampled camera angles are selected in the following order: $(0°, 0°)$, $(0°, \pm20°)$, $(\pm30°, 0°)$, and $(\pm60°, 0°)$. It is worth noting that during actual optimization, camera poses can be arbitrary. This sampling strategy is adopted purely to facilitate a simpler, more consistent, and computationally efficient optimization process. We conduct a total of four optimization rounds. In the first round, we perform inpainting without changing the camera poses, i.e., using poses of the original context views. This step focuses on refining low-confidence regions through inpainting to enhance rendering quality under the original views. In subsequent rounds, we fix the batch size to 4 and include supervision signals from the originally inpainted context views, previously inpainted views, and newly generated inpainted views. For the optimization of Gaussian parameters, we adopt parameter-specific learning rates following the setting proposed in (Qin et al., 2024). The detailed learning rates for each type of parameter are summarized in Table 7. Empirically, we observe that each optimization round converges efficiently within 600 training iterations.

## B.3 OVGO BENCHMARK

For evaluation on our proposed OVGO benchmark, we uniformly sample *16 novel camera poses* around the context image $I_1$, covering a horizontal angular range of $[-60°, 60°]$ and a vertical angular range of $[-20°, 20°]$. To simplify the evaluation setup, we decouple horizontal and vertical

rotations, following the same strategy described in Appendix B.2. The IoU score for every query is computed by averaging over a total of $50 \times 16$ images. If the union of predicted and ground-truth regions in an image is empty, that image is excluded from the IoU computation. To ensure robustness, we repeat the inpainting, growing, and evaluation process five times with the same settings and report the mean IoU as the final benchmark result.

### B.4 RGB-TO-SEMANTIC CONTROLNET MODULE

To ensure spatial alignment between the inpainted RGB image and its corresponding semantic map, we adopt a control mechanism inspired by ControlNet (Zhang et al., 2023), where the RGB image serves as guidance for the generation of the semantic map. An overview of the ControlNet architecture is illustrated in Fig. 8. Specifically, our control module comprises the encoder and bottleneck components of the stable diffusion UNet architecture, with their weights initialized from the corresponding layers of a pretrained stable diffusion UNet. Conditional signals are then injected into the bottleneck and decoder parts via zero convolutions and element-wise addition. To accelerate training and enhance the effectiveness of control learning, we initialize the control module with pretrained parameters from a ControlNet model (Zhang et al., 2023) conditioned on image segmentation. This initialization strategy provides a strong prior for spatially consistent generation and significantly improves both training efficiency and overall performance. Details of the training settings for this module are provided in Table 8.

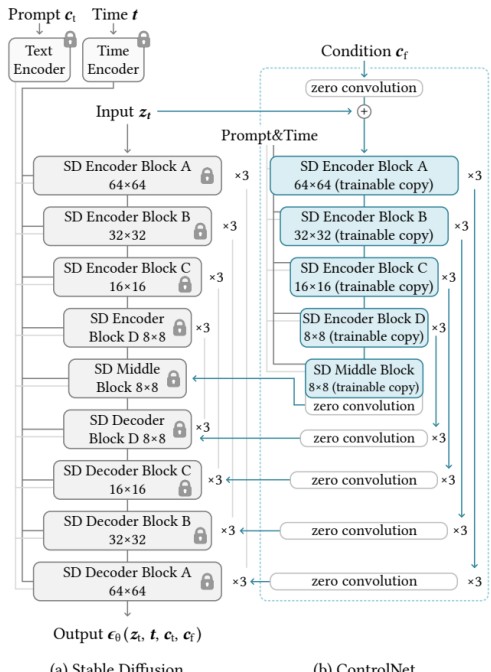

Figure 8: The architecture of the ControlNet (Zhang et al., 2023).

### B.5 RELEVANCE SCORE FOR EVALUATION

During open-vocabulary querying, we select regions with a relevance score greater than 0.5 as the final predicted category mask. The computation of the relevance score is inspired by prior works (Kerr et al., 2023; Qin et al., 2024; Shi et al., 2024), and is defined as follows for each query:

$$\text{Relevance} = \min_i \frac{\exp(g_{\text{img}} \cdot g_{\text{qry}})}{\exp(g_{\text{img}} \cdot g_{\text{qry}}) + \exp(g_{\text{img}} \cdot g_{\text{canon}}^i)}, \tag{10}$$

where $g_{\text{img}}$ denotes the image semantic feature, $g_{\text{qry}}$ is the query APE embedding, and $g_{\text{canon}}^i$ represents the APE embedding of a predefined canonical phrase such as *"object"*, *"things"*, *"stuff"*, or *"texture"*.

In contrast to the mentioned prior works, which typically focus on a limited set of categories in a single scene and require the set of possible scene categories to be known in advance, we adopt a more general strategy. These prior methods often normalize the relevance score and select masks based on a threshold over the normalized values. However, this approach may incorrectly force the prediction of masks even for categories absent in the scene. To address this limitation and enhance generalizability, we directly apply a fixed threshold of 0.5 to the raw (unnormalized) relevance scores and select pixels with scores exceeding this threshold as the final predicted mask. This ensures that only queries with truly high relevance scores produce predictions, avoiding false positives in irrelevant categories. As a result, we are able to compute per-category prediction masks from a predefined query set without requiring manual query specification for each individual scene.

## C  ACKNOWLEDGMENT OF LLM USAGE

We acknowledge the use of Large Language Models (LLMs) in the process of refining certain sections of this manuscript. The LLMs were employed to assist in polishing and improving the clarity of the text, but all content and intellectual contributions remain the result of the authors' research. We emphasize that the LLMs were utilized only as a tool for language enhancement and did not influence the research findings or the overall scientific content presented herein.

