# OpenReview forum: "OGGSplat: Open-Vocabulary Gaussian Growing for Expanded Field-of-View"
_ICLR.cc/2026/Conference — ICLR 2026 Conference Withdrawn Submission_

### Official Review · Reviewer_pJF1 · 2025-10-29

**Soundness:** 3
**Presentation:** 2
**Contribution:** 4
**Rating:** 4
**Confidence:** 3

**Summary:**

This paper addresses a novel problem in feedforward or generalizable novel view synthesis with open-vocabulary semantics and and proposes two key components: (1) RGB-Semantic Consistent Inpaintor that leverages semantic cues around unseen region boundaries through bidirectional RGB–semantic diffusion guidance to inpaint unseen regions reasonably and achieve consistent semantic; (2) a Gaussian Growing Module that progressively integrates the inpainted RGB and semantic maps into the 3D Gaussian representation, enhancing scene completeness and ensuring open-vocabulary consistency across views.

**Strengths:**

1. The idea of converting incomplete semantic cues around unseen region boundaries into textual prompts to guide diffusion-based image inpainting is both novel and well-motivated. This design elegantly leverages the residual semantic information near the boundaries to provide meaningful linguistic guidance, resulting in more semantically coherent and visually plausible inpainting.

2. The paper introduces a new training pipeline that effectively incorporates the newly inpainted RGB and semantic information into the 3D Gaussian representation, enabling gradual scene completion in 3D space.

3.The proposed method achieves competitive or state-of-the-art performance across multiple benchmarks, demonstrating the effectiveness of the overall pipeline design.

**Weaknesses:**

1. The main novelty of the paper lies in the Edge Translator, which leverages incomplete semantic information around region boundaries to generate text prompts that guide the diffusion-based image inpainting. However, the paper lacks ablation studies or in-depth analysis specifically evaluating the effectiveness of this module. Without such evidence, it remains unclear how much this component contributes to the overall improvement.

2. The proposed Gaussian Growing Module inevitably introduces an optimization step into the 3D Gaussian Splatting process, making the overall pipeline no longer fully feedforward.

3. The comparison experiments are somewhat confusing. The chosen baselines do not include semantic completion capabilities, making it difficult to fairly assess the contribution of the proposed pipeline. The authors should compare against 3DGS + diffusion methods such as Difix3D+ [1], GenFusion [2], or See3D [3] under sparse-view settings to more convincingly demonstrate the effectiveness of their approach, since these methods also leverage diffusion priors for scene completion.


[1] Wu, J. Z., Zhang, Y., Turki, H., Ren, X., Gao, J., Shou, M. Z., ... & Ling, H. (2025). Difix3d+: Improving 3d reconstructions with single-step diffusion models. In Proceedings of the Computer Vision and Pattern Recognition Conference (pp. 26024-26035).

[2]Wu, S., Xu, C., Huang, B., Geiger, A., & Chen, A. (2025). Genfusion: Closing the loop between reconstruction and generation via videos. In Proceedings of the Computer Vision and Pattern Recognition Conference (pp. 6078-6088).

[3] Ma, B., Gao, H., Deng, H., Luo, Z., Huang, T., Tang, L., & Wang, X. (2025). You see it, you got it: Learning 3d creation on pose-free videos at scale. In Proceedings of the Computer Vision and Pattern Recognition Conference (pp. 2016-2029).

**Questions:**

1. I am confused about the core problem the paper aims to solve. Is the main goal to improve unseen-region inpainting by leveraging semantic information, or to develop a new approach for predicting unseen regions in open-vocabulary 3D Gaussian Splatting?

2. If it is the second, could the authors elaborate on why unseen-region inpainting is necessary in the open-vocabulary 3DGS setting? How does this approach compare to a 3D generation pipeline that reconstructs an entire scene from a single image and subsequently distills semantic information into 3DGS?

3. Compared to the proposed Edge Translator, what would be the difference if one directly converts the semantic map into text prompts?

---

### Official Review · Reviewer_yWBa · 2025-10-31

**Soundness:** 2
**Presentation:** 3
**Contribution:** 2
**Rating:** 4
**Confidence:** 4

**Summary:**

The paper presents OGGSplat, a pipeline for reconstructing open-vocabulary, semantically aware 3D Gaussian scenes from only two unposed views, with a focus on filling/expanding regions that fall outside the initial view cone. OGGSplat first initializes a generalizable open-vocabulary Gaussian set from two images by Splatt3R with a semantic head distilled from APE features. It then introduces an RGB–semantic consistent inpaintor that runs two diffusion branches (image and semantics) with bidirectional control. The newly inpainted views are lifted to 3D via depth prediction and progressively grown into the Gaussian set with RGB and semantic losses. To measure this, the authors propose the Open-Vocabulary Gaussian Outpainting (OVGO) benchmark on ScanNet++, reporting FID and open-vocabulary mIoU in low-/high-confidence regions (focusing on low-confidence regions that are out of seen views). Experiments show sizable gains over adapted LangSplat and Splatt3R baselines.

**Strengths:**

1. OGGSplat is the first method to generate open-vocabulary 3D Gaussians beyond the input views, a crucial capability for complete scene understanding from sparse data.
2. The introduced OVGO benchmark provides a standardized framework to rigorously evaluate both visual and semantic quality in expanded regions, filling a critical gap in the field.
3. Most ablation studies are detailed and convincingly validate the core design choices, such as the bidirectional inpainting control and the depth alignment strategy.

**Weaknesses:**

1. The paper lacks comparison against a pipeline that uses a 2D/video generative model (e.g., ViewCrafter) for view synthesis followed by 2D segmentation. This is necessary to isolate the specific benefit of the 3D Gaussian representation over a simpler 2D post-processing approach.
2. The baseline comparisons is unfair and incomplete. The current compared methods, LangSplat and Splatt3R, inherently lack any mechanism for hallucinating unseen content. The performance gain shown in Figure 3 stems primarily from the 2D inpainting module, not the 3D optimization. A fairer evaluation would augment the baselines (LangSplat, Splatt3R) with the same inpainting module to directly test the contribution of the proposed 3D Gaussian growing strategy.
3. The semantic backbones lack further investigation. The attribution of the poor *ceiling* performance solely to the APE model is unconvincing, especially since LangSplat (which uses CLIP) performs better on this class. Training OGGSplat with CLIP features is needed to diagnose if the failure is due to APE or the framework itself, and to demonstrate general compatibility with different vision-language models.

**Questions:**

Some questions are not clarified in the manuscript:

1. The Gaussian growing process uses predefined outpainting angles ([-60°, 60°] horizontally, [-20°, 20°] vertically) that are identical to the evaluation benchmark. Does this risk overfit the model to the test distribution? Would results on a more challenging, disjoint set of viewing angles demonstrate stronger generalization?
2. In the ablation study for the Edge Translator (Table 2), the baseline uses a generic text prompt. A more rigorous comparison would be against a strong baseline that uses a detailed, manually crafted prompt describing the entire scene, which is also a common practice in 2D inpainting. Can you show that the Edge Translator outperforms such a scene-level prompt?

---

### Official Review · Reviewer_2Qoz · 2025-11-01

**Soundness:** 2
**Presentation:** 2
**Contribution:** 2
**Rating:** 2
**Confidence:** 5

**Summary:**

The paper presents an *open-vocabulary* 3D Gaussian splatting method that aims to make the *open-vocabulary* 3DGS reconstructions interpolate (and perhaps extrapolate) smoothly between sparse view inputs. The proposed method first initializes the 3D Gaussians from Splatt3R (a few-shot end-to-end regression method for reconstructing 3DGS using DuST3R Transformer). The semantic labels of the Gaussians are first obtained by a vision-language model (APE), and then *outpainted* using ControlNet-augmented diffusion model. The newly introduced semantic outpaints are lifted to 3DGS using depth estimation module. The method is compared with custom benchmarks called OVGO modified from ScanNet++, and shows better quantitative scores than LangSplat and Splatt3R in terms of FID and segmentation IoU.

**Strengths:**

- The problem setting of 3D outpainting for semantic-augmented 3DGS in this paper is novel as authors has claimed.
- The paper contains rich and readable qualitative results.
- Implementation details of the paper is well-addressed and the supplementation material also contains the necessary code to reproduce the results.

**Weaknesses:**

1. Although I acknowledge the novelty of the problem setup, I am not convinced that the problem of outpainting/inpainting semantic-augmented 3DGS, which the paper tries to solve, is a specialized research topic. For example, we may separately apply semantic feature assignment and outpainting/3D reconstruction. Is this separation of problem not feasible?
2. There are only one benchmark used throughout the project, and the proposed method is demonstrated for the proposed benchmarks, which is vulnerable to bias. A suggestion: If the main focus is the novelty of problem, then new comprehensive benchmarks can be marked as the main contribution. If so, various similar methods in the previous paradigm (e.g., general few-shot open-vocabulary 3DGS, including [Gaussian-Grouping](https://arxiv.org/abs/2312.00732), [LEGaussians](https://arxiv.org/pdf/2311.18482), [GOI](https://arxiv.org/pdf/2405.17596), [LangSplatV2](https://openreview.net/forum?id=XR5y4nvTfz)) should be tested for the new benchmark. Alternatively, if the main focus is the novelty of method, then it should be tested for the general benchmarks such as 3D-OVS, Mip-NeRf360, LERF, etc. The current evaluations do not seem to fit in either categories.
3. The paper mixes many existing methods to build a general pipeline. I am not oppose to such “high-level” approaches, but it would be much clearer to have a table, either in the main article or in the appendices, that summarizes the modules adopted and the modules invented, so that we may be able to improve upon this method by improving a submodule in the future works.
4. There are newer methods for *open-vocabulary 3DGS* including [LangSplatV2](https://openreview.net/forum?id=XR5y4nvTfz) (NeurIPS 2025), [GOI](https://arxiv.org/pdf/2405.17596), [LEGaussians](https://arxiv.org/pdf/2311.18482), etc. The latter ones are also cited in the manuscript in lines 116-126. These papers should also be compared in the main experiment sections. Currently, there are only LangSplat (v1) and Splatt3R, which are relatively outdated.

In summary, clearer justification for the problem being solved, unbiased and more comprehensive evaluation, as well as updated comparison results between various previous works are my key suggestions. We can discuss further in the discussion phase to improve the paper.

**Questions:**

I have addressed key concerns in the weaknesses section. Here are some minor points and issues that may be caused by my confusion and misreading.

- How is “Gaussian growing” differs from scene reconstruction from Gaussian splatting? The paper seems to rely on the inpainting/outpainting feature of diffusion model, too, which is typical in 3D scene generation.
- How is applying outpainting semantic Gaussians better than semantically labeling Gaussians from outpainted images?
- How fast Gaussians grow and how long does it take to run for a scene?

---

### Official Review · Reviewer_94Uu · 2025-11-02

**Soundness:** 2
**Presentation:** 2
**Contribution:** 1
**Rating:** 2
**Confidence:** 5

**Summary:**

The paper introduces OGGSplat, a method for 3D Gaussian scene reconstruction from sparse views with the focus on Gaussians enriched with semantic (open-vocabulary) features (from [1]). Since feed-forward, sparse-view 3D reconstruction approaches leveraging 3D Gaussians are usually limited to the field of view of the input views, i.e., Gaussians are only placed within input camera frustums, this paper proposes to extrapolate by inpainting both rendered RGB and feature maps and iteratively grow 3D Gaussians in these extrapolated novel views.
For this, the authors propose an "RGB-semantic consistent inpaintor", where semantic features at boundaries of incomplete renderings are transformed into text prompts for RGB inpainting with a pre-trained text-to-image diffusion model and a ControlNet architecture is used to inpaint semantic features conditioned on RGB.
During an iterative 3D Gaussian growing procedure, inpainted novel views as supervision for newly grown Gaussians are progressively integrated.
The paper further proposes an "Open-Vocabulary Gaussian Outpainting" benchmark including view selection and evaluation procedure on a validation subset of ScanNet++, which the authors use to compare their method with one optimization-based and one feed-forward baseline.
Finally, the paper includes qualitative results for zero-shot generalization on another dataset (S3DIS) and an ablation study w.r.t. conditioning of the inpainting module.

References:
- [1] Aligning and Prompting Everything All at Once for Universal Visual Perception. CVPR 2024

**Strengths:**

- The paper is mostly well written and easy to follow and understand.
  - The introduction motivates the importance of semantically meaningful 3D representations well for applications in embodied AI, for example.
  - The related work covers important works distilling semantic features into 3D Gaussians via optimization and feed-forward prediction.
  - The method section is well structured following the order of the method: Gaussian initialization, RGB and feature inpainting, and Gaussian growing.
  - The Sec. 4.1 about the Open-Vocabulary Gaussian Outpainting benchmark provides important details for reproducibility.
- The paper includes an ablation study to evaluate the effect of the conditionings during inpainting.
- The appendix provides further experimental results including ablations and comprehensive implementation details.
- The supplementary material provides code as well as videos showcasing novel view renderings and semantic querying of the 3D scene.

**Weaknesses:**

- The paper ignores all existing generative 3D reconstruction approaches completely, both in related work and as baselines:
  - Feed-forward generative 3D scene reconstruction / extrapolation methods are among others: [1, 2, 3, 4]
  - Optimization-based generative 3D scene reconstruction approaches are for example: [5, 6, 7, 8, 9]
- The method is limited in terms of novelty, technical soundness, and clarity:
  - The paper combines generative 3D reconstruction using pre-trained diffusion models with semantic 3D Gaussians. Both individually are not novel.
    - There are many methods for generative sparse-view 3D scene reconstruction with 3D Gaussians as representation as listed above, but unfortunately neglected by the authors.
    - Adding semantic features, e.g., from vision-language models to 3D Gaussians for language-based (open-vocabulary) segmentation of 3D scenes is also not novel by itself, but the authors do report related works regarding this and choose LangSplat as a baseline.
  - Both technical contributions of this paper are of limited novelty: RGB-semantic consistent inpaintor (Sec. 3.2) and open-vocab. Gaussian Growing (Sec. 3.3).
    - The RGB-semantic consistent inpaintor consists of semantic feature to RGB inpainting and RGB to semantic feature inpainting.
      - For the semantic feature to RGB inpainting, the authors convert semantic features at boundaries of novel view renderings to a text prompt for a T2I diffusion model.
        - For this, they leverage the top 100 semantic categories of the training dataset, which should limit the generalization and makes the use of the term "open-vocabulary" questionable.
        - Furthermore, converting the semantic boundary features into a manually designed text prompt listing all training categories matching the features is a quite crude way of conditioning in my opinion. A simple and more precise conditioning could be using the rendered partial semantic feature map directly, e.g., using a ControlNet just like for the semantic feature inpainting conditioned on RGB as well.
        - The authors do ablate on this design choice, but choose a generic prompt "a room" as conditioning for the T2I model as baseline, which is the most naive and weak alternative possible, rendering this ablation very uninformative.
      - For the other way around (RGB to semantic features), I do not understand the need for a diffusion-based inpainting model for semantic features in the first place. Why should that be better than just applying the feature extractor (in this case APE [10]) on the inpainted RGB novel views?
        - The paper lacks motivation for why this model is necessary at all.
        - It also lacks detailed information about how it is trained, whether it is a pixel-space or latent-space model, and if latent space, whether the authors train a new autoencoder for these semantic features.
        - The authors do provide an ablation study comparing with an "offline open-vocab. sem. segmentation model, e.g., SAM+CLIP", but this is qualitative only (Fig. 5) and figure, baseline, and description of results lack clarity. Tbh, I did not understand this part at all.
    - Regarding, the open-vocabulary Gaussian Growing, I do not see any technical novelty.
      - Everything from view selection, to initialization using depth estimators, over progressive optimization strategy, and loss terms has been done by previous works for optimization-based generative 3D reconstruction [5, 6, 7, 8] in at least similar fashions.
      - The only difference is the additional feature loss to incorporate the inpainted feature maps as supervision for the feature part of the open-vocabulary Gaussians.
- The experimental evaluation and comparison with baselines is too limited:
  - The paper includes only two baselines that are both too weak and a comparison with them is not fair:
    - The authors modify Splatt3R by adding a semantic head, which is actually a sub-component of their method, as they also rely on this modified Splatt3R for initialization. Therefore, this can also be seen as an ablation. Anyway, this baseline cannot do any extrapolation such that novel view renderings are just incomplete with black areas resulting in bad FID and IoU scores obviously.
    - LangSplat also does not incorporate any generative prior for scene completion such that it simply overfits to the sparse views and then has the same black regions + additional blur.
    - The opacity threshold (0.01) (line 361) is too low such that evaluation will be done on the incomplete novel view renderings of baselines.
  - The proposed benchmark considers only a validation subset of a single dataset: ScanNet++
    - While the authors do provide zero-shot generalization results on another dataset S3DIS, these are also similar office scenes as in ScanNet, the results are only qualitative, and they do not include any baselines.
  - As mentioned above already, the ablation study regarding conditionings includes too weak baselines:
    - For RGB inpainting, the baseline is a T2I model with the generic "a room" prompt. A stronger baseline would be direct conditioning on incomplete feature maps, or even more comparable with existing approaches would be a multi-view or nowadays camera-conditioned video diffusion model.
    - For the feature inpainting, the baseline is just leaving out the RGB conditioning, if I am not mistaken. Here, it remains unclear why we cannot just use APE on the inpainted RGB images.
- The related work limits itself to 3D Gaussian splatting, but for a more comprehensive overview, the authors should briefly mention the lines of works based on NeRFs.
- The paper lacks some technical details:
  - Is only the semantic head trained for Splatt3R or do you fine-tune end-to-end?
  - What architecture do you use for the autoencoder of APE features to compress the channel dimension. Is it trained separately or end-to-end with the modified Splatt3R?
  - In line 229f., the authors state that "categories are encoded". How exactly? Are these just learnable embeddings per category?
  - In line 252f., how are the (novel) anchor views selected?

References:
- [1] latentSplat: Autoencoding Variational Gaussians for Fast Generalizable 3D Reconstruction. ECCV 2024
- [2] MVSplat360: Feed-Forward 360 Scene Synthesis from Sparse Views. NeurIPS 2024
- [3] Bolt3D: Generating 3D Scenes in Seconds. ICCV 2025
- [4] Wonderland: Navigating 3D Scenes from a Single Image. CVPR 2025
- [5] DiffusioNeRF: Regularizing Neural Radiance Fields with Denoising Diffusion Models. CVPR 2023
- [6] ReconFusion: 3D Reconstruction with Diffusion Priors. CVPR 2024
- [7] CAT3D: Create Anything in 3D with Multi-View Diffusion Models. NeurIPS 2024
- [8] Sp2360: Sparse-view 360 Scene Reconstruction using Cascaded 2D Diffusion Priors. ECCV Wild3D 2024
- [9] ViewCrafter: Taming Video Diffusion Models for High-fidelity Novel View Synthesis. TPAMI 2025
- [10] Aligning and Prompting Everything All at Once for Universal Visual Perception. CVPR 2024

**Questions:**

For a rebuttal, the only suggestions I can give are a significantly more extensive evaluation and comparison with fair baselines that leverage generative priors for 3D scene reconstruction. For example, a simple combination of some of these works like ViewCrafter with LangSplat and semantic supervision from feature maps directly predicted with APE on inpainted novel views would be a much better baseline in terms of fairness in my opinion, but probably also much harder to outperform.
Furthermore, as described in the weaknesses, the baseline selection in the ablation would need to be improved.
There are several open questions regarding technical details about the APE feature autoencoder, the semantic Splatt3R head, the category encoding, the novel view selection and the diffusion model for feature inpainting.

However, I have to say that even if the authors follow these suggestions in the rebuttal, the concerns about the limited novelty would remain, unfortunately.

---

### Note · Authors · 2025-11-13

I have read and agree with the venue's withdrawal policy on behalf of myself and my co-authors.